# A New Family of Generalization Bounds Using Samplewise Evaluated CMI

**Fredrik Hellström**
Chalmers University of Technology
Gothenburg, Sweden
`frehells@chalmers.se`

**Giuseppe Durisi**
Chalmers University of Technology
Gothenburg, Sweden
`durisi@chalmers.se`

## Abstract

We present a new family of information-theoretic generalization bounds, in which the training loss and the population loss are compared through a jointly convex function. This function is upper-bounded in terms of the disintegrated, sample-wise, evaluated conditional mutual information (CMI), an information measure that depends on the losses incurred by the selected hypothesis, rather than on the hypothesis itself, as is common in probably approximately correct (PAC)-Bayesian results. We demonstrate the generality of this framework by recovering and extending previously known information-theoretic bounds. Furthermore, using the evaluated CMI, we derive a samplewise, average version of Seeger's PAC-Bayesian bound, where the convex function is the binary KL divergence. In some scenarios, this novel bound results in a tighter characterization of the population loss of deep neural networks than previous bounds. Finally, we derive high-probability versions of some of these average bounds. We demonstrate the unifying nature of the evaluated CMI bounds by using them to recover average and high-probability generalization bounds for multiclass classification with finite Natarajan dimension.

## 1   Introduction

Information-theoretic generalization bounds, i.e., generalization bounds that are expressed in terms of information-theoretic metrics such as the Kullback-Leibler (KL) divergence, mutual information, and conditional mutual information (CMI), have emerged as useful tools to obtain an accurate characterization of the performance of deep neural networks. To obtain such bounds, one compares a *posterior*, that is, the distribution over the hypotheses induced by the learning algorithm, to a reference distribution called the *prior*, an approach first introduced to provide probably approximately correct (PAC) bounds for Bayesian classifiers [1, 2, 3, 4]. The connection between these results and classic information-theoretic metrics was clarified in [5, 6], where the generalization gap, averaged with respect to the joint posterior and data distribution, is bounded in terms of the mutual information between the training data and the hypothesis. These bounds have since been extended in several ways [7, 8, 9, 10, 11, 12, 13, 14, 15].

A major step was taken in [16], in which a setting where the training set is randomly selected from a larger supersample is considered. We refer to this setup as the CMI setting, as it leads to generalization bounds in terms of the CMI between the hypothesis and training set selection, given the supersample. It turns out that these bounds can be further tightened by observing that the selected hypothesis enters the derivation only through the loss that it induces on the supersample. Using this observation, [16] also derives bounds in terms of the information encoded in these losses rather than in the hypothesis itself, a quantity referred to as the *evaluated CMI* (e-CMI). Due to the data-processing inequality, these bounds are always tighter than the regular CMI bounds. This observation was recently further

36th Conference on Neural Information Processing Systems (NeurIPS 2022).

exploited by [17], which derived samplewise versions of these bounds.[1] Intriguingly, these bounds are both easier to evaluate than their hypothesis-based counterparts, due to the lower dimensionality of the random variables involved, and quantitatively tighter for deep neural networks. In particular, while bounds involving information measures based on the hypothesis space tend to increase as training progresses [11, 18], the bound of [17] remains stable.

The results that have so far been derived for the CMI setting pertain only to the (weighted) difference between population loss and training loss, or to its squared or absolute value [8, 9, 14, 16, 17, 18]. In the PAC-Bayesian literature, other types of discrepancy measures have been considered, and shown to result in tighter bounds on the population loss. For example, [19, 20, 21, 22] consider the binary KL divergence (i.e., the KL divergence between two Bernoulli distributions) with parameters given by the training and population loss, respectively, while [23, 24] consider arbitrary jointly convex functions. Finally, [25] allows for arbitrary functions. It should be noted that in all of these results, a moment-generating function that depends on the selected function has to be controlled for the bound to be computable.

Recently, the e-CMI framework was proven to be expressive enough to allow one to rederive known results in learning theory, e.g., generalization bounds expressed in terms of algorithmic stability, VC dimension, and related complexity measures [17, 26]. Tightening and extending e-CMI bounds, which is the main objective of this paper, has the potential to further increase the unifying nature of the e-CMI framework.

**Contributions**  Leveraging a basic inequality involving a generic convex function of two random variables (Lemma 1), we establish several novel disintegrated, samplewise, e-CMI bounds on the average generalization error. Specifically, we present i) a square-root bound (Theorem 1) on the generalization error, together with a mean-squared error extension, which tightens the bound recently reported in [17]; ii) a linear bound (Theorem 3) that tightens the bound given in [18]; iii) a binary KL bound (Theorem 4), which is a natural extension to the e-CMI setting of a well-known bound in the PAC-Bayes literature [22]. While the derivation of the first two bounds involves an adaptation of results available in the literature, to obtain the binary KL bound we need a novel concentration inequality involving independent but not identically distributed random variables (Lemma 2). As an additional contribution, we show how to adapt the techniques presented in the paper to obtain high-probability (rather than average) e-CMI bounds (Theorem 7). Furthermore, we illustrate the expressiveness of the e-CMI framework by using our bounds to recover average and high-probability generalization bounds for multiclass classification with finite Natarajan dimension (Theorem 8). Finally, we conduct numerical experiments on MNIST and CIFAR10, which reveal that the binary KL bound results in a tighter characterization of the population loss compared to the square-root bound and linear bound for some deep learning scenarios.

**Preliminaries and Notation**

We let $D(P \,||\, Q)$ denote the KL divergence between the two probability measures $P$ and $Q$. This quantity is well-defined if $P$ is absolutely continuous with respect to $Q$. When $P$ and $Q$ are Bernoulli distributions with parameters $p$ and $q$ respectively, we let $D(P \,||\, Q) = d(p \,||\, q) = p \log(p/q) + (1 - p) \log((1 - p)/(1 - q))$, and we refer to $d(p \,||\, q)$ as the binary KL divergence. For two random variables $X$ and $Y$ with joint distribution $P_{XY}$ and respective marginals $P_X$ and $P_Y$, we let $I(X; Y) = D(P_{XY} \,||\, P_X P_Y)$ denote their mutual information. Throughout the paper, we use uppercase letters to denote random variables and lowercase letters to denote their realizations. We denote the conditional joint distribution of $X$ and $Y$ given an instance $Z = z$ by $P_{XY|Z=z}$ and the corresponding conditional distribution for the product of marginals by $P_{X|Z=z}P_{Y|Z=z}$. Furthermore, we let $I^z(X; Y) = D(P_{XY|Z=z} \,||\, P_{X|Z=z}P_{Y|Z=z})$, which is referred to as the *disintegrated* mutual information [11]. Its expectation is the conditional mutual information $I(X; Y|Z) = \mathbb{E}_Z\big[I^Z(X; Y)\big]$.

**CMI Setting**  Let $\mathcal{Z}$ be the sample space and let $\mathcal{D}$ denote the data-generating distribution. Consider a supersample $\widetilde{Z} \in \mathcal{Z}^{n \times 2}$, where each entry is generated independently from $\mathcal{D}$. For convenience, we index the columns of $\widetilde{Z}$ starting from 0 and the rows starting from 1. Furthermore, we denote the $i$th

---

[1]While [17] considers the information stored in predictions rather than losses, referred to as the $f$-CMI, the derivations therein can be adapted to obtain bounds that depend on the losses, as we clarify in Section 2.2.

row of $\widetilde{Z}$ as $\widetilde{Z}_i$. Let $S = (S_1, \ldots, S_n)$ denote a membership vector, with entries generated independently according to a $\mathrm{Bern}(1/2)$ distribution, independently of $\widetilde{Z}$. Let $\bar{S} = (1 - S_1, \ldots, 1 - S_n)$ denote the modulo-2 complement of $S$. We refer to $S$ as a membership vector because it is used to divide the supersample into an $n$-dimensional training vector $\widetilde{Z}_S$ with entries $[\widetilde{Z}_S]_i = \widetilde{Z}_{i,S_i}$ and an $n$-dimensional test vector $\widetilde{Z}_{\bar{S}}$ with entries $[\widetilde{Z}_{\bar{S}}]_i = \widetilde{Z}_{i,\bar{S}_i}$. To be able to handle arbitrary learning settings, we consider learning algorithms as maps $\mathcal{A} : \mathcal{Z}^n \times \mathcal{R} \to \mathcal{F}$, where $R \in \mathcal{R}$ is a random variable (independent of $\widetilde{Z}$ and $S$) that captures the stochasticity of the algorithm and $\mathcal{F}$ is a space, for instance, a parameter space or function space. For a fixed $R = r$ and $\widetilde{Z}_S = \tilde{z}_s$, $\mathcal{A}(\tilde{z}_s, r)$ is a deterministic function of $\tilde{z}_s$. The quality of the learning algorithm's output is evaluated using a bounded loss function $\ell : \mathcal{F} \times \mathcal{Z} \to [0, 1]$. We let $U$ be a vector of size $m$, the elements of which are sampled without replacement uniformly at random from $(1, \ldots, n)$. For a given realization $U = u = (u_1, \ldots, u_m)$, we let $\tilde{z}_u$ denote the $m \times 2$ matrix obtained by stacking the vectors $\tilde{z}_{u_i}$ for $i = 1, \ldots, m$. Similarly, $\ell(\mathcal{A}(\tilde{z}_s, R), \tilde{z}_u)$ denotes the $m \times 2$ matrix of losses obtained by applying $\ell(\mathcal{A}(\tilde{z}_s, R), \cdot)$ elementwise to $\tilde{z}_u$. We denote the population loss as $L_{\mathcal{D}}(\mathcal{A}, \tilde{z}_s, r) = \mathbb{E}_{Z'}[\ell(\mathcal{A}(\tilde{z}_s, r), Z')]$, where $Z' \sim \mathcal{D}$. The training loss is $L_{\tilde{z}_s}(\mathcal{A}, \tilde{z}_s, r) = \frac{1}{n} \sum_{i=1}^{n} \ell(\mathcal{A}(\tilde{z}_s, r), [\tilde{z}_s]_i)$. In general, for any $p$-dimensional vector of data points $\hat{z}$, we let $L_{\hat{z}}(\mathcal{A}, \tilde{z}_s, r) = \frac{1}{p} \sum_{i=1}^{p} \ell(\mathcal{A}(\tilde{z}_s, r), \hat{z}_i)$. Since $\mathcal{A}(\tilde{z}_s, r)$ does not depend on $\tilde{z}_{\bar{s}}$, $L_{\tilde{z}_{\bar{s}}}(\mathcal{A}, \tilde{z}_s, r)$ is a test loss. Finally, we let $L_{\mathcal{D}} = \mathbb{E}_{\widetilde{Z}, S, R}[L_{\mathcal{D}}(\mathcal{A}, \widetilde{Z}_S, R)]$ denote the average population loss and $\hat{L} = \mathbb{E}_{\widetilde{Z}, S, R}[L_{\widetilde{Z}_S}(\mathcal{A}, \widetilde{Z}_S, R)]$ denote the average training loss.

## 2 Average Generalization Bounds

In this section, we present the main results of this paper: a new family of disintegrated samplewise e-CMI bounds for the average population loss. High-probability versions of these bounds are given in Section 3.

### 2.1 Main Lemma

We now present the generic inequality upon which the bounds in this section are based. This inequality, which is similar in nature to the ones provided in [23] and [25], gives us a generic framework to derive generalization bounds, as it allows for a wide choice of functions for comparing training loss and population loss. A crucial difference compared to [23] and [25] is that our focus in this section is on average rather than PAC-Bayesian generalization bounds, and on bounds based on the disintegrated, samplewise e-CMI, rather than traditional KL-based bounds.

**Lemma 1.** *Let $f_\gamma : [0, 1]^2 \to \mathbb{R}$ be a function that is jointly convex in its arguments and is parameterized by $\gamma$. Let $X$ and $Y$ be two random variables, and let $Y'$ be a random variable with the same marginal distribution as $Y$ such that $Y'$ and $X$ are independent. Assume that the joint distribution of $X, Y$ is absolutely continuous with respect to the joint distribution of $X, Y'$. Let $g_1(X, Y)$ and $g_2(X, Y)$ be measurable functions with range $[0, 1]$ and finite first moments, such that, for all $\gamma$, $\mathbb{E}_{X,Y}[f_\gamma(g_1(X, Y), g_2(X, Y))]$ is finite. Let*

$$\xi_\gamma = \log \mathbb{E}_{X,Y'}\left[ e^{f_\gamma\left(g_1(X,Y'), g_2(X,Y')\right)} \right] \tag{1}$$

*and assume that $\xi_\gamma$ is finite. Then,*

$$\sup_\gamma f_\gamma(\mathbb{E}_{X,Y}[g_1(X, Y)], \mathbb{E}_{X,Y}[g_2(X, Y)]) - \xi_\gamma \leq \sup_\gamma \mathbb{E}_{X,Y}[f_\gamma(g_1(X, Y), g_2(X, Y))] - \xi_\gamma$$

$$\leq I(X; Y). \tag{2}$$

The proof, which is an application of Jensen's inequality and Donsker-Varadhan's variational representation of the KL divergence, is given in Appendix A, along with the proofs of all other results provided in this paper.

Note that we are allowed to optimize $\gamma$ because, in this section, we only consider average bounds. In contrast, when we derive high-probability bounds in Section 3, we need to fix $\gamma$. Optimizing $\gamma$ over a set of candidate values in the high-probability scenario incurs a union bound cost [27, Sec. 1.2.2].

To apply Lemma 1, we need to identify functions $f_\gamma(\cdot, \cdot)$ for which the moment generating function in (1) can be controlled. A theme throughout this section is that we identify functions $f_\gamma(\cdot, \cdot)$ for

which there exist concentration results that imply that $\xi_\gamma \leq 0$. This allows us to loosen the bound in (2) by discarding $\xi_\gamma$.

Throughout this section, we assume that the loss is bounded, so that $\ell(\cdot, \cdot) \in [0, 1]$. Note that the reported results can be extended to more general losses by use of scaling. In particular, assume that there is a function $K : \mathcal{F} \to \mathbb{R}_+$ such that, for all $f \in \mathcal{F}$, $\sup_z \ell(f, z) \leq K(f)$ (referred to as the hypothesis-dependent range condition in [28]). Then, all of the results that we present for bounded losses also hold for the scaled loss $\ell(f, z)/K(f)$.

## 2.2 Extending ($f$)-CMI Bounds to e-CMI

We now apply Lemma 1 to recover the average bounds of [17, 18]. These works derive bounds using the information contained either in the hypothesis itself or the resulting predictions, i.e., the CMI or the $f$-CMI. We instead derive bounds in terms of the information captured by the matrix of losses, i.e., the e-CMI. For parametric supervised learning algorithms, we can recover the original bounds via the data-processing inequality. This is formalized in the following remark.

**Remark 1.** *Consider a parametric supervised learning setting, where $\mathcal{Z} = \mathcal{X} \times \mathcal{Y}$ and $\mathcal{A}(\tilde{z}_S, R) = W \in \mathcal{F}$ are the parameters of a function $\phi_W : \mathcal{X} \to \mathcal{Y}$. Let $\tilde{x}_u$ denote the $m \times 2$ matrix obtained by projecting each element of $\tilde{z}_u$ onto $\mathcal{X}$, i.e., the matrix of unlabeled examples. Let $\phi_W(\tilde{x}_u)$ denote the matrix of predictions obtained by elementwise application of $\phi_W$ to $\tilde{x}_u$. Then, for any fixed $\tilde{z}$ and $u$,*

$$I^{\tilde{z},u}(\ell(W, \tilde{z}_u); S_u) \leq I^{\tilde{z},u}(\phi_W(\tilde{x}_u); S_u) \leq I^{\tilde{z},u}(W; S_u). \tag{3}$$

We now use Lemma 1 to recover some of the results reported in [17]. Let $\Delta = L_{\tilde{z}_{S'_i}}(\mathcal{A}, \tilde{z}_s, R) - L_{\tilde{z}_{S'_i}}(\mathcal{A}, \tilde{z}_s, R)$, where $S'_i$ is an independent copy of $S_i$. For a fixed $\tilde{z}$, symmetry implies that $\mathbb{E}_{S'_i}[\Delta] = 0$. Furthermore, since $\ell(\cdot, \cdot) \in [0, 1]$, we have that $\Delta \in [-1, 1]$. Thus, $\Delta$ is 1-sub-Gaussian. By using properties of sub-Gaussian random variables, we can control $\xi_\gamma$ in Lemma 1 when choosing $f_\gamma(\cdot, \cdot)$ as $\gamma\Delta$ or $\gamma\Delta^2$. The resulting bounds are given in the following theorem.

**Theorem 1** (Square-root bound and squared bound). *Consider the CMI setting. Then,*

$$\left| L_\mathcal{D} - \hat{L} \right| \leq \frac{1}{n} \sum_{i=1}^{n} \mathbb{E}_{\widetilde{Z}}\left[ \sqrt{2I^{\widetilde{Z}}(\ell(\mathcal{A}(\widetilde{Z}_S, R), \widetilde{Z}_i); S_i)} \right] \tag{4}$$

$$\leq \frac{1}{n} \sum_{i=1}^{n} \sqrt{2I(\ell(\mathcal{A}(\widetilde{Z}_S, R), \widetilde{Z}_i); S_i | \widetilde{Z})}. \tag{5}$$

*Furthermore,*

$$\mathbb{E}_{\widetilde{Z}, R, S}\left[ \left( L_\mathcal{D}(\mathcal{A}, \widetilde{Z}_S, R) - L_{\widetilde{Z}_S}(\mathcal{A}, \widetilde{Z}_S, R) \right)^2 \right] \leq \frac{8}{m}\left( I(\ell(\mathcal{A}(\widetilde{Z}_S, R), \widetilde{Z}_U); S_U | \widetilde{Z}, U) + 2 \right). \tag{6}$$

In (4), the expectation over $\widetilde{Z}$ is taken outside of the square root, and the generalization bound is in terms of the disintegrated mutual information. By Jensen's inequality, this is tighter than (5).

As shown in Appendix A, one can further generalize (4) to obtain an upper bound of the form

$$\mathbb{E}_{\widetilde{Z}, U}\left[ \sqrt{2I^{\widetilde{Z},U}(\ell(\mathcal{A}(\widetilde{Z}_S, R), \widetilde{Z}_U); S_U)} \right]. \tag{7}$$

However, since (7) increases with the size $m$ of the random subset $U$ [17, Prop. 1], we focus only on the case $m = 1$, as this leads to the tightest bound. The same holds for (5), as well as for all of the bounds that we present in the remainder of this section, where we also focus only on $m = 1$. In contrast, the choice of $m = 1$ in the squared bound in (6) is suboptimal and actually yields a vacuous bound due to the $2/m$ term.

It is possible to obtain a bound similar to (4), but where an expectation over $R$ is taken outside the square root and the mutual information is also conditioned on $R$. We present this bound in the following theorem.

**Theorem 2** ($R$-conditioned square-root bound). *Consider the CMI setting. Then,*

$$\left| L_{\mathcal{D}} - \hat{L} \right| \le \frac{1}{n} \sum_{i=1}^{n} \mathbb{E}_{\widetilde{Z},R} \left[ \sqrt{2 I^{\widetilde{Z},R}(\ell(\mathcal{A}(\widetilde{Z}_S, R), \widetilde{Z}_i); S_i)} \right] \tag{8}$$

$$\le \frac{1}{n} \sum_{i=1}^{n} \sqrt{2 I(\ell(\mathcal{A}(\widetilde{Z}_S, R), \widetilde{Z}_i); S_i | \widetilde{Z}, R)}. \tag{9}$$

A similar bound, but given in terms of mutual information rather than e-CMI, is reported in [11, Thm. 2.4]. The randomness of the learning algorithm can reduce the mutual information between its output and the selection variable $S_i$. Specifically, one can show that

$$I^{\tilde{z}}(\ell(\mathcal{A}(\tilde{z}_S, R), \tilde{z}_i); S_i) \le I^{\tilde{z}}(\ell(\mathcal{A}(\tilde{z}_S, R), \tilde{z}_i); S_i | R) \tag{10}$$

which implies that the square-root bound in (5) is tighter than the $R$-conditioned square-root bound in (9). However, the ordering between the disintegrated square-root bound in (4) and the disintegrated $R$-conditioned square-root bound in (8) is unclear.

Next, we generalize the linear bounds of [18], which are samplewise extensions of the bounds in [16], to the e-CMI framework. We consider two scenarios. For the first one, we assume that $\gamma = (\gamma_1, \gamma_2)$ are positive constants that satisfy a certain constraint, and let $f_\gamma(\hat{L}, L_{\mathcal{D}}) = \gamma_1 n (L_{\mathcal{D}} - \gamma_2 \hat{L})$. For the second one, we assume that $\hat{L} = 0$, the so-called *interpolating* setting, and let $f_\gamma(\hat{L}, L_{\mathcal{D}}) = n \log(2) L_{\mathcal{D}}$. For both of these scenarios, it can be shown that $\xi_\gamma \le 0$, yielding the following.

**Theorem 3** (Linear bound and interpolation bound). *Consider the CMI setting. Let $\Gamma \subset \mathbb{R}_+^2$ denote the set of parameters $\gamma = (\gamma_1, \gamma_2)$ that satisfy $\gamma_1(1 - \gamma_2) + (e^{\gamma_1} - 1 - \gamma_1)(1 + \gamma_2^2) \le 0$. Then,*

$$L_{\mathcal{D}} \le \min_{\gamma \in \Gamma} \gamma_2 \hat{L} + \sum_{i=1}^{n} \frac{I(\ell(\mathcal{A}(\widetilde{Z}_S, R), \widetilde{Z}_i); S_i | \widetilde{Z})}{\gamma_1 n}. \tag{11}$$

*Furthermore, if $\hat{L} = 0$,*

$$L_{\mathcal{D}} \le \sum_{i=1}^{n} \frac{I(\ell(\mathcal{A}(\widetilde{Z}_S, R), \widetilde{Z}_i); S_i | \widetilde{Z})}{n \log(2)}. \tag{12}$$

The interpolation bound in (12) improves on the linear bound in (11) when $\hat{L} = 0$, as the constraint implies that $\gamma_1^2 - 4(e^{\gamma_1} - 1)(e^{\gamma_1} - 1 - \gamma_1) \ge 0$. This means that $\gamma_1 < 0.37$, whereas $\log 2 > 0.69$.

### 2.3 Binary KL Bound with Samplewise e-CMI

We now derive bounds in terms of the binary KL divergence between the training loss and the test loss. To this end, similar to [27, 29], we define

$$d_\gamma(q \,||\, p) = \gamma q - \log(1 - p + p e^\gamma). \tag{13}$$

An important property of this function is that

$$\sup_\gamma d_\gamma(q \,||\, p) = d(q \,||\, p). \tag{14}$$

Note that both $d_\gamma(\cdot \,||\, \cdot)$ and $d(\cdot \,||\, \cdot)$ are jointly convex in their arguments. For our next result, we need the following lemma.

**Lemma 2.** *For $i = 1, \ldots, n$, let $X_i \sim P_{X_i}$, $\mathbb{E}[X_i] = \mu_i$, $\hat{\mu} = \frac{1}{n} \sum_{i=1}^{n} X_i$, and $\bar{\mu} = \frac{1}{n} \sum_{i=1}^{n} \mu_i$. Assume that $X_i \in [0,1]$ almost surely and that all $X_i$ are independent. Then, for every fixed $\gamma > 0$,*

$$\mathbb{E}[\exp(n d_\gamma(\hat{\mu} \,||\, \bar{\mu}))] \le 1. \tag{15}$$

This inequality, which, to the best of our knowledge, has previously been reported only for identically distributed random variables, follows by combining [30, Lemma 1], [31, Thm. 3], and [29, Eq. (17)] (which is a generalization of [27, Lemma 1.1.1] from binary to bounded random variables).[2]

---

[2] A similar result, with $d(\cdot \,||\, \cdot)$ instead of $d_\gamma(\cdot \,||\, \cdot)$, is established in [32, Lemma 2].

Consider a fixed $\tilde{z}$. For each $i$, $\mathbb{E}_{S_i'}\left[L_{[\tilde{z}_{S'}]_i}(\mathcal{A}, \tilde{z}_s, R)\right] = (\ell(\mathcal{A}(\tilde{z}_s, R), \tilde{z}_{i,0}) + \ell(\mathcal{A}(\tilde{z}_s, R), \tilde{z}_{i,1}))/2$, where $S_i'$ is an independent copy of $S_i$. Thus,

$$\mathbb{E}_{S'}\left[L_{\tilde{z}_{S'}}(\mathcal{A}, \tilde{z}_s, R)\right] = \frac{L_{\tilde{z}_s}(\mathcal{A}, \tilde{z}_s, R) + L_{\tilde{z}_{\bar{s}}}(\mathcal{A}, \tilde{z}_s, R)}{2}. \tag{16}$$

This observation allows us to bound the binary KL divergence between the training loss $\hat{L}$ and the arithmetic mean of the training and population loss, $(\hat{L} + L_{\mathcal{D}})/2$. Combining (15) and (2), with appropriate choices for the variables and functions therein, we obtain the following result.

**Theorem 4** (Binary KL bound). *Consider the CMI setting. Then,*

$$d\left(\hat{L} \,\|\, \frac{L_{\mathcal{D}} + \hat{L}}{2}\right) \leq \frac{1}{n}\sum_{i=1}^{n} I(\ell(\mathcal{A}(\widetilde{Z}_S, R), \widetilde{Z}_i); S_i | \widetilde{Z}) \tag{17}$$

*which implies that $L_{\mathcal{D}}$ can be bounded as*

$$L_{\mathcal{D}} \leq d^{-1}\left(\hat{L}, \frac{1}{n}\sum_{i=1}^{n} I(\ell(\mathcal{A}(\widetilde{Z}_S, R), \widetilde{Z}_i); S_i | \widetilde{Z})\right) \tag{18}$$

*where*

$$d^{-1}(q, c) = \sup\left\{p \in [0,1] : d\left(q \,\|\, \frac{q+p}{2}\right) \leq c\right\}. \tag{19}$$

Similar to Theorem 1, we can obtain a disintegrated version of (18) where the expectation over $\widetilde{Z}$ is outside of the inversion of the binary KL divergence. We present this result in the following theorem.

**Theorem 5** (Disintegrated binary KL bound). *Consider the CMI setting. Then,*

$$L_{\mathcal{D}} \leq \mathbb{E}_{\widetilde{Z}}\left[d^{-1}\left(\mathbb{E}_{R,S}\left[L_{\widetilde{Z}_S}(\mathcal{A}, \widetilde{Z}_S, R)\right], \frac{1}{n}\sum_{i=1}^{n} I^{\widetilde{Z}}(\ell(\mathcal{A}(\widetilde{Z}_S, R), \widetilde{Z}_i); S_i)\right)\right]. \tag{20}$$

By using Pinsker's inequality, which implies that $2(q-p)^2 \leq d(q\,\|\,p)$, one can weaken (18) to obtain

$$\left|L_{\mathcal{D}} - \hat{L}\right| \leq \sqrt{\frac{2}{n}\sum_{i=1}^{n} I(\ell(\mathcal{A}(\widetilde{Z}_S, R), \widetilde{Z}_i); S_i | \widetilde{Z})}. \tag{21}$$

However, since the average over $i$ is inside the square root, this bound is weaker than (5) by Jensen's inequality.

To establish Theorem 4, it is crucial that the supremum over $\gamma$ in Lemma 1 is outside the expectation that defines $\xi_\gamma$. Indeed, $\sup_\gamma d_\gamma(q\,\|\,p) = d(q\,\|\,p)$, and with the notation from Lemma 2, we have [22, Thm. 1]

$$\mathbb{E}[\exp(nd(\hat{\mu}\,\|\,\bar{\mu}))] \geq \sqrt{n}. \tag{22}$$

Therefore, having the supremum inside of the expectation would unavoidably lead to an additional term greater than $\log(\sqrt{n})/n$ in the upper bound of (18).

The binary KL bound in (18) is an average, samplewise, e-CMI analogue of the PAC-Bayesian bound with a binary KL divergence on the left-hand side, sometimes referred to as Seeger's bound [4, 20]. However, as noted after Lemma 2, the concentration inequality needed to establish Theorem 4 is a refinement of the result used in [20]. Also, the dependence on $\hat{L}$ and $L_{\mathcal{D}}$ in the binary KL bound is nonstandard. As we show in Section 5, the binary KL bound in (18) is sometimes tighter than the square-root bound in Theorem 1 and the linear bound in Theorem 3.

The binary KL bounds in (18) and (20) can be tightened by considering affine transformations of the arguments of $d(\cdot\,\|\,\cdot)$. However, this does not lead to improvements for the low training loss scenarios that we focus on in this paper. Therefore, we relegate these extensions to Appendix B, where we also present an analogue of Theorem 4 in terms of the samplewise mutual information between the learning algorithm's output and the training data, rather than the e-CMI.

While $d^{-1}(\cdot, \cdot)$ does not admit an analytical expression, $d^{-1}(0, \cdot)$ is tractable. This leads to the following simplified form of the disintegrated binary KL bound in (20) when $\hat{L} = 0$.

**Theorem 6** (Disintegrated interpolation binary KL bound). *Assume that $\hat{L} = 0$. Then, (20) becomes*

$$L_{\mathcal{D}} \leq \mathbb{E}_{\widetilde{Z}}\left[2 - 2e^{-\frac{1}{n}\sum_{i=1}^{n} I^{\widetilde{Z}}(\ell(\mathcal{A}(\widetilde{Z}_S, R), \widetilde{Z}_i); S_i)}\right]. \tag{23}$$

# 3 High-Probability Bounds

The techniques presented in Section 2 can be adapted to allow for the derivation of high-probability bounds, in particular through the use of exponential inequalities [33, 34, 35]. To demonstrate this, we now present two such high-probability bounds.

**Theorem 7** (High-probability square-root and binary KL bounds)**.** *Let $\lambda = \ell(\mathcal{A}(\widetilde{Z}_S, R), \widetilde{Z})$. Furthermore, let $P_{\lambda|\widetilde{Z}S}$ denote the conditional distribution of $\lambda$ given $\widetilde{Z}$ and $S$, and let $P_{\lambda|\widetilde{Z}}$ denote the conditional distribution of $\lambda$ given $\widetilde{Z}$. Then, with probability at least $1 - \delta$ over the draw of $\widetilde{Z}$ and $S$,*

$$\mathbb{E}_R\Big[L_{\widetilde{Z}_{\bar{S}}}(\mathcal{A}, \widetilde{Z}_S, R) - L_{\widetilde{Z}_S}(\mathcal{A}, \widetilde{Z}_S, R)\Big] \leq \sqrt{\frac{2}{n-1}\bigg(D(P_{\lambda|\widetilde{Z}S} \,||\, P_{\lambda|\widetilde{Z}}) + \log \frac{\sqrt{n}}{\delta}\bigg)}. \tag{24}$$

*Furthermore, also with probability at least $1 - \delta$ over the draw of $\widetilde{Z}$ and $S$,*

$$d\Bigg(\mathbb{E}_R\Big[L_{\widetilde{Z}_S}(\mathcal{A}, \widetilde{Z}_S, R)\Big] \,||\, \mathbb{E}_R\bigg[\frac{L_{\widetilde{Z}_S}(\mathcal{A}, \widetilde{Z}_S, R) + L_{\widetilde{Z}_{\bar{S}}}(\mathcal{A}, \widetilde{Z}_S, R)}{2}\bigg]\Bigg)$$
$$\leq \frac{D(P_{\lambda|\widetilde{Z}S} \,||\, P_{\lambda|\widetilde{Z}}) + \log \frac{2\sqrt{n}}{\delta}}{n}. \tag{25}$$

These high-probability bounds involve the empirical test loss of the learner, and not the population loss. However, a simple use of the triangle inequality, as detailed in [34, Thm. 3], allows one to make (24) and (25) explicit in the population loss. Note that the information measures that appear in the bounds depend on the full training set, rather than individual samples. As proven in [36], samplewise versions of these tail bounds are typically loose. For instance, the tail bound with samplewise information measures in [37, Lemma 8] contains a constant term and has a detrimental linear dependence on $1/\delta$, instead of the benign logarithmic dependence in (24) and (25). Finally, under a stronger technical assumption of absolute continuity, one can obtain high-probability bounds not only with respect to the draw of $\widetilde{Z}$ and $S$, but also the randomness $R$ of the learning algorithm. We present this result in Appendix B.

# 4 Expressiveness of the e-CMI Framework

We now illustrate the unifying nature of the e-CMI framework by demonstrating its expressiveness. Specifically, proceeding similarly to [17, 26], we use the e-CMI framework to rederive known bounds in learning theory. In particular, we consider multiclass classification with finite Natarajan dimension. In the next theorem, we provide a bound on the e-CMI appearing in the results reported in Section 2, as well as a bound on the data-dependent KL divergence appearing in (24) and (25).

**Theorem 8.** *Consider a multiclass classification setting, for which $\mathcal{Z} = \mathcal{X} \times \mathcal{Y}$, where $\mathcal{X}$ is the instance space and $\mathcal{Y}$ the label space, and assume that $|\mathcal{Y}| = N$. Furthermore, assume that the learning algorithm implements a function $f : \mathcal{X} \to \mathcal{Y}$ where $f \in \mathcal{F}$ belongs to a class of finite Natarajan dimension $d_N$ [38]. Finally, assume that $2n > d_N + 1$. Then,*

$$I(\ell(\mathcal{A}(\widetilde{Z}_S, R), \widetilde{Z}); S|\widetilde{Z}) \leq d_N \log\bigg(\binom{N}{2}\frac{2en}{d_N}\bigg). \tag{26}$$

*Furthermore, with probability at least $1 - \delta$ under the draw of $\widetilde{Z}$ and $S$,*

$$D(P_{\lambda|\widetilde{Z}S} \,||\, P_{\lambda|\widetilde{Z}}) \leq d_N \log\bigg(\binom{N}{2}\frac{2en}{d_N}\bigg) + \log\frac{1}{\delta}. \tag{27}$$

To obtain generalization bounds from Theorem 8, we need to upper-bound the information terms that appear in the bounds of Sections 2 and 3 by using either (26) or (27). While this can be done for any of the information-theoretic generalization bounds in this paper, we present two concrete examples in the following corollary.

**Corollary 1.** *Consider a multiclass classification setting, for which $\mathcal{Z} = \mathcal{X} \times \mathcal{Y}$, where $\mathcal{X}$ is the instance space and $\mathcal{Y}$ the label space, and assume that $|\mathcal{Y}| = N$. Furthermore, assume that the learning algorithm implements a function $f : \mathcal{X} \to \mathcal{Y}$ where $f \in \mathcal{F}$ belongs to a class of finite Natarajan dimension $d_N$ [38]. Finally, assume that $2n \geq d_N + 1$. Then,*

$$\left| L_{\mathcal{D}} - \hat{L} \right| \leq \sqrt{\frac{2 d_N \log\left(\binom{N}{2} \frac{2en}{d_N}\right)}{n}}. \tag{28}$$

*Furthermore, with probability at least $1 - \delta$ under the draw of $\widetilde{Z}$ and $S$,*

$$d\left( \mathbb{E}_R\left[ L_{\widetilde{Z}_S}(\mathcal{A}, \widetilde{Z}_S, R) \right] \; \| \; \mathbb{E}_R\left[ \frac{L_{\widetilde{Z}_S}(\mathcal{A}, \widetilde{Z}_S, R) + L_{\widetilde{Z}_{\bar{S}}}(\mathcal{A}, \widetilde{Z}_S, R)}{2} \right] \right)$$
$$\leq \frac{d_N \log\left(\binom{N}{2} \frac{2en}{d_N}\right) + \log \frac{2}{\delta} + \log \frac{4\sqrt{n}}{\delta}}{n}. \tag{29}$$

The scaling behavior of (28) with respect to $n$ recovers the standard rate found in the literature [39]. This is minimax optimal up to a logarithmic term [40, Thm. 29.3].

## 5 Comparing the Bounds

We now perform some comparisons between the bounds obtained in Section 2. As it turns out, there is no clear ordering between the different bounds in general. For a first comparison, we consider the interpolating setting, where the training loss is zero. We focus on the case where the size $m$ of the random subset $U$ equals 1. Since the squared bound in (6) is vacuous for this choice, it is not considered for this comparison. Moreover, we assume that the learning algorithm is, on average, indifferent to permutations of the training set. Specifically, we assume that the value of $I(\ell(\mathcal{A}(\widetilde{Z}_S, R), \widetilde{Z}_i); S_i | \widetilde{Z})$ equals a constant $B$ that does not depend on the index $i$. For a more straightforward comparison, we also exclude the disintegrated bounds in (4) and (20), and postpone their evaluation to Section 6. Thus, in this section, we compare the square-root bound in (5), the linear bound in (11), the interpolation bound in (12), and the binary KL bound in (18). In the following proposition, we establish an ordering between these bounds. The result follows by straightforward arithmetic.

**Proposition 1.** *Assume that $I(\ell(\mathcal{A}(\widetilde{Z}_S, R), \widetilde{Z}_i); S_i | \widetilde{Z}) = B$ for all $i$ and that $\hat{L} = 0$. Let $\gamma_{1,\mathrm{opt}}$ be the largest $\gamma_1$ such that $\gamma_1^2 - 4(e^{\gamma_1} - 1)(e^{\gamma_1} - 1 - \gamma_1) \geq 0$, and assume that $B < 2\gamma_{1,\mathrm{opt}}^2 \approx 0.27$. Then, the upper bounds on $L_{\mathcal{D}}$ are, in increasing order, the interpolation bound in (12), the binary KL bound in (18), the linear bound in (11), and the square-root bound in (5). If $B > 2\gamma_{1,\mathrm{opt}}^2$, the ordering between the square-root bound in (11) and the linear bound in (5) is inverted.*

While Proposition 1 gives the ordering between the bounds for the interpolating scenario, the quantitative difference between them is not clear. A numerical illustration is given in Figure 1a, under the same assumptions as in Proposition 1. As established in Proposition 1, the interpolation bound in (12) is superior across the range of values for $B$, and the binary KL bound in (18) is slightly looser. However, while the constant $\log 2$ in (12) is sharp, the derivation is valid only for interpolating learning algorithms. In contrast, the other bounds are only marginally affected when we allow for a small, non-zero training loss. This added flexibility shows the usefulness of (18) as compared to (12).

To obtain a more complete picture, we numerically evaluate the bounds for a range of values for $\hat{L}$ and $B$. This is illustrated in Figure 1b. While the binary KL bound in (18) is tightest for most small values of $B$ and $\hat{L}$, a region of this space is dominated by the linear bound in (11). When both $B$ and $\hat{L}$ grow larger, the square-root bound in (5) is the tightest. Thus, the picture that emerges from these comparisons is that, when the training loss is zero, it is best to use (12), while we need (18) for the case of a small, but non-zero, training loss. However, if the training loss is high, one needs to evaluate all three bounds and take the minimum.

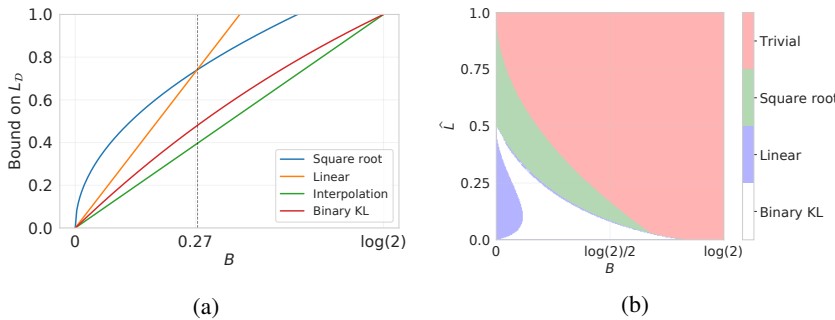

(a)                (b)

Figure 1: (a): A quantitative comparison between the square-root bound in (5), the linear bound in (11), the interpolation bound in (12), and the binary KL bound in (18) under the assumptions of Proposition 1. (b): A comparison between the square-root bound in (5), the linear bound in (11) and the binary KL bound in (18) under the assumptions of Proposition 1, but with non-zero $\hat{L}$. Each point in the figure is color-coded according to which bound gives the tightest characterization of $L_{\mathcal{D}}$ for the given parameters. For the region labeled *Trivial*, no bound performs better than the trivial $L_{\mathcal{D}} \leq 1$.

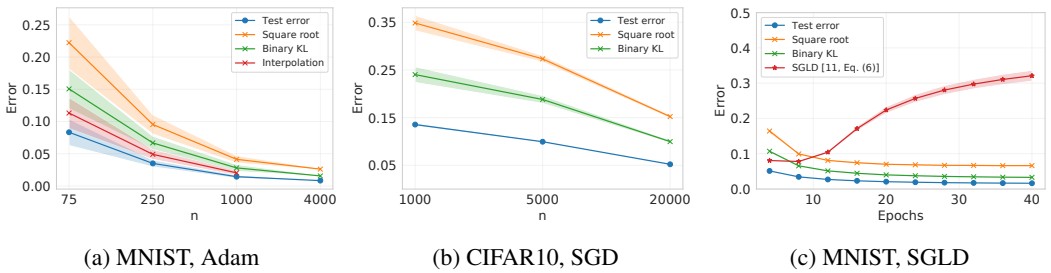

(a) MNIST, Adam       (b) CIFAR10, SGD      (c) MNIST, SGLD

Figure 2: Numerical evaluation of the test error in three deep learning scenarios, along with the upper bounds provided by the square-root bound in (4), the binary KL bound in (20) and, when applicable, the interpolation bound in (12) and the SGLD bound in [11, Eq. 6].

## 6  Numerical Results

We now consider three deep learning settings and evaluate the bounds derived in Section 2. To the best of our knowledge, the tightest average generalization bounds available in the literature for typical deep learning scenarios, such as CNNs trained on MNIST [41] or CIFAR10 [42], are found in [17]. These numerical results are based on [17, Eq. 22], which is a version of (4) where the e-CMI is replaced by the f-CMI. In order to perform a direct comparison, we consider the same experimental setups. The code for our experiments is largely based on the code from [17].[3] Additional numerical results for a setting with randomized labels are given in Appendix C. A detailed description of the training methods, network architectures, and experimental setup is given in Appendix D.

The previous section indicated that the interpolation bound in (12) and the binary KL bound in (20) are the superior bounds when the training loss is zero or low, respectively, which is common for deep learning. Hence, we focus on these bounds, as well as on the square-root bound in (4). When evaluating the bounds, we use the classification error as the loss function. We consider two learning algorithms, namely stochastic gradient descent (SGD) with a fixed random seed and stochastic gradient Langevin dynamics (SGLD). The first is a deterministic learning algorithm with high relevance for current practice. For deterministic learning algorithms, generalization bounds based on the information captured by the weights of a neural network are vacuous, but as noted by [17], bounds based on the information contained in the predictions (or the losses they induce) are typically nonvacuous. The second is a randomized learning algorithm, which allows for comparison to weight-based information-theoretic generalization bounds. For this setting, we also evaluate the SGLD bound of [11, Eq. 6].

---

[3]Available: `https://github.com/hrayrhar/f-CMI`.

First, we consider a CNN trained with Adam (a variant of SGD) on a binarized version of MNIST, where we only consider the digits $4$ and $9$. The results are reported in Figure 2a. In binary classification, there is a one-to-one mapping between predictions and losses. Hence, for this scenario, the information captured by the matrix of losses is the same as the information captured by the matrix of predictions. For this scenario, the binary KL bound in (20) significantly improves on the square-root bound in (4). For $n = 75$, $250$, and $1000$, all of the neural networks that we trained achieved zero training error, making the interpolation bound in (12) a valid bound. This bound results in the tightest characterization of the test error. However, for $n = 4000$, the training error was non-zero for some neural networks. As a consequence, the interpolation bound in (12) was not applicable.

Next, in Figure 2b, we look at ResNet-50 pretrained on ImageNet and fine-tuned using SGD on CIFAR10. In multi-class classification, the map from predictions to losses is no longer one-to-one. Thus, this is a setting where the e-CMI is potentially lower than the f-CMI. Again, the binary KL bound in (20) significantly improves on the square-root bound in (4).

Finally, as an example of a randomized learning algorithm, we consider a CNN trained with SGLD on the binarized MNIST data set. This is shown in Figure 2c. As training progresses, both (20) and (4) improve on the SGLD bound in [11, Eq. 6]. However, the latter is lower for the initial epochs. While (20) reduces the magnitude of this discrepancy, which was also observed in [17], it does not fully close the gap. This may partially be due to the fact that the bound in [11, Eq. 6] has an expectation over $R$ outside of the square root, similar to (8), which may be beneficial during early epochs. Another possibility is that the value that we compute for the mutual information may be an overestimate, especially for low values of the true mutual information. This is an effect of the non-negativity of the mutual information.

# 7 Discussion and Limitations

In this paper, we presented several generalization bounds in terms of the evaluated CMI. While the results hold for a generically formulated learning problem, they are derived under the assumption of a bounded loss function and independent and identically distributed (i.i.d.) training data. While the boundedness assumption can be alleviated to some degree by using the same type of argument as in [16, Thm. 5.1], it is unclear to what extent the i.i.d. assumption can be relaxed.

Our experiments demonstrate that some of the derived bounds are numerically accurate for several deep learning scenarios. As shown in Appendix C, this remains true also when one considers randomized labels and varies the width, depth, and learning rate of the neural network. These results indicate that this family of bounds is potentially powerful enough to guide the design of deep neural networks. This is an intriguing avenue for future research.

## Acknowledgements

We thank Hrayr Harutyunyan for helpful comments regarding the numerical evaluation. This work was partly supported by the Wallenberg AI, Autonomous Systems and Software Program (WASP) funded by the Knut and Alice Wallenberg Foundation and the Chalmers AI Research Center (CHAIR).

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
