# OpenReview forum: "A New Family of Generalization Bounds Using Samplewise Evaluated CMI"
_NeurIPS.cc/2022/Conference — NeurIPS 2022 Accept_

### Official Review · Reviewer_4bSW · 2022-07-09

**Rating:** 5
**Confidence:** 4
**Soundness:** 3 good
**Presentation:** 3 good
**Contribution:** 2 fair

**Summary:**

This paper considers the problem of providing generalization bounds based on the information-theoretic measures. In particular, this paper is on extending and improving the work by Steinke-Zakynthinou and Harutyunyan et al. One line of work in this area is on proving generalization bounds based on “sample-wise” measures, initiated by Bu et al. The main contribution of this work is proving new generalization bounds by combining the idea of evaluated CMI and sample-wise generalization bounds. The authors also provide some generalization bounds based on “disintegrated mutual information” formalized by Negrea et al. One generalization bound which is new in this line of work is the binary KL bounds in Section 2.3. Furthermore, the authors provide some numerical results showing that their bounds are not vacuous in DL settings.



**Questions:**

I have raised some concerns and asked several questions in the Strengths And Weaknesses part.

**Limitations:**

I could not find any section on the potential limitations in the paper.

**Strengths And Weaknesses:**

The most interesting aspect of the paper is providing a generic tool in Lemma 1 which can be used to redervie existing bounds, and proving new results. Also, the results on the binary KL bounds are completely novel, and have not been reported before.

My main concern about this paper is novelty. It seems the authors combine, in a straightforward way, a line of work by Steinke-Zakynthinou, Harutyunyan et al., and Bu et al. Also, in terms of technical novelty, I found the proof of Lemma 1 straightforward. I would like the authors to clarify on the novelty of their work.

-How can we compare the high probability bounds in Section 3 with the results of Grünwald-Steinke-Zakynthinou, Hellström-Durisi? Are the bounds in the paper tighter?

-Is the result in Theorem 8 minmax optimal?

-I would like to raise a general concern regarding the numerical results. In general, in learning theory we use the generalization frameworks to gain new insights regarding the algorithms. For instance, the existing results for generalization of SGD show that the size of the weights can determine the generalization performance; for SGLD, we know the covariance of the training samples’ gradients affects the generalization, etc. I would like to compare this approach with the case that we are “only” interested in the “value” of the generalization error. In the former case, we often use the validation set, and the numbers we report can accurately estimate the test error. First question is what can we learn about your numerical results? From the experiment details, I think you need the holdout samples to estimate the mutual information.  If you use the holdout samples to estimate the test error, what would be the results? In general, what is the gain of using your bounds compared to the validation set approach?

---

> ### Author Response · Authors · 2022-08-02
> **Response to Reviewer 4bSW, part 1**
>
> We thank the reviewer for their constructive comments.
>
> ___________________________________________________________
> > _My main concern about this paper is novelty. It seems the authors combine, in a straightforward way, a line of work by Steinke-Zakynthinou, Harutyunyan et al., and Bu et al. Also, in terms of technical novelty, I found the proof of Lemma 1 straightforward. I would like the authors to clarify on the novelty of their work._
>
> As the reviewer points out, the results in this paper largely build on powerful techniques available in the literature.
> However, from a technical perspective, the bounds that are given in terms of the binary KL divergence rely on the concentration inequality in Lemma 2, which pertains to non-identically distributed random variables and is, to the best of our knowledge, novel.
> Using this new concentration result, we avoid a logarithmic dependence on $n$ that would appear if we used standard techniques.
> Furthermore, previously obtained PAC-Bayesian results in terms of the binary KL divergence have not been derived for the e-CMI setting.
> Deriving such bounds for the e-CMI setting, which we do in this paper, requires careful handling of the random variables involved, leading to the unexpected form of Theorem 4 that Reviewer CMvk points out.
> This is in contrast to bounds based on the ordinary MI, which we discuss and extend in Appendix B.1.
> Finally, regarding the expressiveness of the e-CMI framework, previous work considers average bounds, while we take extra steps to bound the pointwise information measures that appear in our high-probability bounds.
> Thus, while Lemma 1 and its proof are straightforward, applying Lemma 1 for the specific convex functions and settings that we consider is non-trivial.
> Finally, the resulting bounds demonstrably improve upon previous work for some cases.
>
> ___________________________________________________________
> > _-How can we compare the high probability bounds in Section 3 with the results of Grünwald-Steinke-Zakynthinou, Hellström-Durisi? Are the bounds in the paper tighter?_
>
> Due to the use of evaluated CMI rather than the ordinary CMI that appears in (Grünwald, Steinke and Zakynthinou, 2021) and (Hellström and Durisi, 2021), the results in this paper can potentially be significantly tighter.
> By the data-processing inequality for KL divergence [39, Thm. 2.2.6], our bounds can be relaxed to obtain bounds with the weight-based KL divergence in place of the evaluated KL divergence, demonstrating that they compare favorably to those of (Hellström and Durisi, 2021).
> In terms of convergence rates, the result of (Grünwald, Steinke and Zakynthinou, 2021) interpolates between our square-root and linear bounds, where the specific rate depends on the parameter $\beta^*$ in their Bernstein condition.
> For the case of the $0/1$-loss that we consider, we have that $B=4$ and $\beta^*=0$ in their Bernstein condition.
> Thus, they obtain the $1/\sqrt n$ rate of our square-root bound.
> Quantitatively, one expects our square-root bound to be tighter for several reasons:
> 1. their bounds include a constant $1/\eta_{\textnormal{max}}>28.8$ that multiplies the KL divergence and $\log 1/\delta$ terms,
> 2. their bound includes an extra $\eta_{\textnormal{max}}/(4\sqrt n)$ term, and
> 3. the $\log 1/\delta$ term in their bound appears outside of the square-root containing the KL divergence.
>
> To perform a direct quantitative comparison, some simplifying assumptions are needed.
> First, while the data-processing inequality implies that the KL divergence in Eq. (24) of our paper is always smaller than or equal to the KL divergence in the bound of (Corollary 1, Grünwald, Steinke and Zakynthinou, 2021), we shall assume that both KL divergences equal $1$.
> Furthermore, we set $\delta=0.01$ and $n=1000$.
> Under these assumptions, the bound in (Corollary 1, Grünwald, Steinke and Zakynthinou, 2021) gives a generalization gap of approximately $2.89$, which is vacuous, whereas the bound in Eq. (24) of our paper gives a generalization gap of approximately $0.13$.
> This discrepancy arises mainly due to the large constants described above, and holds for other reasonable values of the parameters involved.
> In the revised version of the paper, we will include a more detailed comparison.
>
> ___________________________________________________________
> > _-Is the result in Theorem 8 minmax optimal?_
>
> The generalization bounds in App. B.4, Eq. (117-118), and thus the bounds on the information terms in Theorem 8, are minimax optimal up to the logarithmic dependence on $n$, which can be removed for some cases (see, e.g., “*Understanding Machine Learning: From Theory to Algorithms*”, Shai Shalev-Shwartz and Shai Ben-David, Theorem 29.3).
> Our bounds can be weakened to obtain these known bounds (up to a logarithmic term).
> We will clarify this in the revised version of the paper.

---

> > ### Author Response · Authors · 2022-08-02
> > **Response to Reviewer 4bSW, part 2**
> >
> > ___________________________________________________________
> > > _-I would like to raise a general concern regarding the numerical results. In general, in learning theory we use the generalization frameworks to gain new insights regarding the algorithms. For instance, the existing results for generalization of SGD show that the size of the weights can determine the generalization performance; for SGLD, we know the covariance of the training samples’ gradients affects the generalization, etc. I would like to compare this approach with the case that we are “only” interested in the “value” of the generalization error. In the former case, we often use the validation set, and the numbers we report can accurately estimate the test error. First question is what can we learn about your numerical results? From the experiment details, I think you need the holdout samples to estimate the mutual information. If you use the holdout samples to estimate the test error, what would be the results? In general, what is the gain of using your bounds compared to the validation set approach?_
> >
> > While previous work on SGD has shown that the size of the weights can provide upper bounds on the generalization gap, the tightness of these bounds, and thus their explanatory power, is an empirical question.
> > To the best of our knowledge, norm-based bounds are typically vacuous for deep neural networks, indicating that another mechanism for generalization is at play.
> > Drawing insights from vacuous bounds can be misleading: if the bounds are loose, the explanation they offer for generalization may be irrelevant for the generalization performance that is actually observed.
> >
> > The purpose of our numerical experiments is to probe the explanatory power of our bounds.
> > In this paper, we present upper bounds on the generalization gap in terms of the information that the losses of the learning algorithm on both training and test samples reveal about the selected data points in the CMI setting, given the entire supersample.
> > Our numerical results indicate that this quantity does capture the bulk of generalization performance for the settings that we consider.
> > Establishing this is the main goal of our empirical evaluations: if a bound closely follows the true test error---as we find that our bounds do for the settings that we study, which include experiments with partly randomized labels---it must capture the mechanism for generalization that is actually at play in the given setting.
> >
> > As we show in Section 4, the e-CMI perspective is expressive enough to recover generalization guarantees in terms of classical complexity measures like the Natarajan dimension, and as shown by (Harutyunyan et al., 2021), it also captures some notions of algorithmic stability.
> > Thus, while the validation set approach is the natural way to proceed in order to numerically estimate the test error---indeed, it is what we compare our bounds to in Fig. 2---the purpose of our numerical experiments is to probe the explanatory power of our bounds.
> > Further pursuing this line of inquiry to determine the concrete mechanism by which SGD-trained neural networks achieve small e-CMI is an interesting area for further study, but beyond the scope of this paper.
> >
> > Based on the suggestion of Reviewer CMvk, we have also extended our numerical experiments to see whether the correlation between our bounds and the test error holds when the width, depth, and learning rate for the network are varied.
> > The results, which indicate that our bounds do display such correlation, can be found in Figure 5 of Appendix C in the revised supplementary material.
> >
> > We will add a section entitled "Limitations" where we will highlight what can be learned from our numerical experiments, namely that the good agreement between our bounds and the true test error, which holds for several tasks, learning algorithms, and even randomized labels, indicates that this family of bounds are potentially powerful enough to guide the design of deep neural networks, while acknowledging that this paper does not provide any recipe for how to do so.

---

> > > ### Comment · Reviewer_4bSW · 2022-08-07
> > > **Thank you.**
> > >
> > > I would like to thank the authors for their detailed response.
> > > Recently, there is a new proposal for using cmi-based technique. The idea has been proposed in the following two paper
> > > -https://arxiv.org/abs/2207.00581
> > > -https://arxiv.org/abs/2206.14800
> > >
> > > Is there any connection between your work and the aforementioned papers? I think it bears similarity with your sample-wise approach.
> > >
> > > I will raise my score to 5.

---

> > > > ### Author Response · Authors · 2022-08-08
> > > > **Thank you**
> > > >
> > > > Thank you for your detailed review, your helpful comments and suggestions, and for raising the score. Also, thank you for pointing out these two interesting papers. These two papers, which were posted after the NeurIPS submission deadline, seem highly related to our samplewise approach. However, since they are very recent, we do not yet know the precise connection.

---

### Official Review · Reviewer_PHyo · 2022-07-11

**Rating:** 7
**Confidence:** 3
**Soundness:** 4 excellent
**Presentation:** 3 good
**Contribution:** 3 good

**Summary:**

This paper derives new information-theoretic generalization bounds based on sample-wise evaluated CMI, among which the square-root bound and the linear bound are tighter than existing bounds, and the binary KL bound is empirically shown to be superior for low-training loss settings. Furthermore, they also show that the derived bounds can be used to recover generalization bounds for multiclass classification with finite Natarajan dimension.

**Questions:**

1. How tight are the bounds in Theorem 8? Is it possible to have some plots demonstrating how much the improvement is?
2. It may be worth mentioning in previous sections that some terms in the derived bounds will later be shown to have upper bounds in Section 4.

**Limitations:**

The assumptions made for the theorems to hold are stated clearly.

**Strengths And Weaknesses:**

Strengths:
This paper is rich, presenting a series of results both novel and deep. The presentation is complete in the sense that discussions on the comparison of the bounds and the corresponding numerical demonstration are included.

Weaknesses:
The notation is heavy, which is necessary, but makes it a bit hard to follow all the results. Maybe adding a table that concludes the derived bounds in Section 5 will make the comparison part more clear.

---

> ### Author Response · Authors · 2022-08-02
> **Response to Reviewer PHyo**
>
> We thank the reviewer for their constructive comments.
>
> ___________________________________________________________
> > _Weaknesses: The notation is heavy, which is necessary, but makes it a bit hard to follow all the results. Maybe adding a table that concludes the derived bounds in Section 5 will make the comparison part more clear._
>
> If the reviewer found any specific parts of the results or the notation to be particularly heavy, we will make an extra effort to rectify this.
> In the revised version of the paper, we will begin Section 5 with a table or other form of summary to clarify the comparison part.
>
> ___________________________________________________________
> > _Questions: How tight are the bounds in Theorem 8? Is it possible to have some plots demonstrating how much the improvement is?_
>
> The generalization bounds in App. B.4, Eq. (117-118), and thus the bounds on the information terms in Theorem 8, are minimax optimal up to a logarithmic dependence on $n$, which can be removed for some cases (see, e.g., “*Understanding Machine Learning: From Theory to Algorithms*”, Shai Shalev-Shwartz and Shai Ben-David, Theorem 29.3).
> Thus, the main point of App. B.4 and Theorem 8 is to show that our information-theoretic bounds lead to (up to a logarithmic term) the same convergence rates as known, minimax optimal bounds.
> We will clarify this in the revised version of the paper.
>
> ___________________________________________________________
> > _It may be worth mentioning in previous sections that some terms in the derived bounds will later be shown to have upper bounds in Section 4._
>
> We thank the reviewer for their helpful suggestion, which we will implement in the revised version of the paper.

---

> > ### Comment · Reviewer_PHyo · 2022-08-09
> > **Response to rebuttal**
> >
> > Thank you for the response. I will maintain my rating.

---

> > > ### Author Response · Authors · 2022-08-09
> > > **Thank you**
> > >
> > > Thank you for your detailed review, your helpful comments and suggestions, and for responding to the rebuttal.

---

### Official Review · Reviewer_kjBx · 2022-07-12

**Rating:** 7
**Confidence:** 4
**Soundness:** 3 good
**Presentation:** 3 good
**Contribution:** 3 good

**Summary:**

This paper presents a new family of information-theoretic generalization bounds in terms of the disintegrated, sample-wise, evaluated conditional mutual information (CMI), an information measure that depends on the losses incurred by the selected hypothesis rather than on the hypothesis itself. The authors demonstrate the generality of this framework by recovering and extending previously known information-theoretic bounds. In some scenarios, this novel bound results in a tighter characterization of the population loss of deep neural networks than previous bounds.


**Questions:**

I am curious if the proposed e-CMI bound could provide a tight characterization of the generalization for the Gaussian mean estimation example considered in section IV-A of [10]? If it works, it would be a good example for readers to compare different types of bounds and understand the idea.

I have one minor comment about the organization. The high-probability bound in section 3 seems to be standard in this type of analysis, and the usage of e-CMI is more critical here. Maybe the Gaussian example and the current Section 4 (Appendix B.3) should be highlighted?

--------------------------------
Post rebuttal:
The explanation of the Gaussian example resolves my concern. It is a nice paper. I will keep my score unchanged.




**Limitations:**

I think the authors fully addressed the limitations and potential negative societal impact of their work.

**Strengths And Weaknesses:**

This paper tightens existing information-theoretic generalization error bound using a simple but effective idea, i.e., the information measure involves the losses incurred by the selected hypothesis is always smaller than that of the hypothesis itself. Remark 1 clearly states this benefit using data processing inequality. Although a similar idea of using f-CMI has been adopted in [15], such an improvement significantly enlarges the usage of information-theoretic generalization error bound as it can be evaluated numerically for deep neural networks, as shown in Sections 5 and 6.

The paper is well-written and technically sound to me.

---

> ### Author Response · Authors · 2022-08-02
> **Response to Reviewer kjBx**
>
> We thank the reviewer for their constructive comments.
>
> ___________________________________________________________
>  > _Questions: I am curious if the proposed e-CMI bound could provide a tight characterization of the generalization for the Gaussian mean estimation example considered in section IV-A of [10]? If it works, it would be a good example for readers to compare different types of bounds and understand the idea._
>
> This is an intriguing question.
> It seems to us that e-CMI does not lead to any gains for this setting compared to previous work.
> The reason is as follows.
> Given the values of the two samples in $ \widetilde{Z}_i$ and the corresponding losses that they induce, we can determine the estimated mean, since we know its distance to two points and that it is a real number.
> Thus, the e-CMI coincides with the f-CMI.
> Since the prediction in this case is the full output of the learning algorithm, the f-CMI also coincides with the regular CMI.
> Thus, the result obtained from our square-root bound coincides with that of (Corollary 1, " _Individually conditional individual mutual information bound on generalization error_", Ruida Zhou, Chao Tian, and Tie Liu, 2021).
> The specific application of this bound to Gaussian mean estimation is given in (Prop. 2, " _Individually conditional individual mutual information bound on generalization error_", Ruida Zhou, Chao Tian, and Tie Liu, 2021).
>
> ___________________________________________________________
>  > _I have one minor comment about the organization. The high-probability bound in section 3 seems to be standard in this type of analysis, and the usage of e-CMI is more critical here. Maybe the Gaussian example and the current Section 4 (Appendix B.3) should be highlighted?_
>
> We thank the reviewer for their proposed improvements.
> In the revised version of the paper, we will highlight Section 4 to a greater extent to accommodate these suggestions.
> While it cannot be done during the rebuttal and discussion phase due to the 9 page limit, we intend to move parts of Appendix B.3 to the main paper when the limit is increased to 10 pages.

---

### Official Review · Reviewer_CMvk · 2022-07-12

**Rating:** 7
**Confidence:** 4
**Soundness:** 3 good
**Presentation:** 3 good
**Contribution:** 2 fair

**Summary:**

PAC-Bayes generalization bounds are often derived from Donsker Varadhan, allowing one to bound a function g that depends on the empirical risk and risk of the predictor. Choosing different forms of g gives rise to different bounds, that may be tighter or loser depending on the regime (often empirical risk magnitude). When it comes to information theoretic bounds, these bounds often come in the most “basic” form, that is comparable to taking g to be of a form of risk - empirical risk. Part of the contribution of the paper is deriving a diverse set of bounds that depend on information-theoretic quantities, similarly as it’s been done in PAC-Bayes setting. The authors present both, bounds that hold in expectations, and high probability bounds.

**Questions:**

The comment under Theorem 8 says that the “resulting bounds essentially recover previously established generalization guarantees”. Are these previously established guarantees optimal? Are there lower bounds? What does “recover” here mean? Is the rate wrt the number of samples optimal in the minimax sense?

Numerical results (Section 6):
- the authors do not really explain what they hope to gain by performing the empirical evaluations, and what questions they are answering. It seems that the motivation is evaluating the numerical tightness of the bounds. Is that right? It would have been nice to extend the evaluation in a few dimensions, beyond varying the number of training epochs and the number of training points. I encourage the authors to check out a paper by Dziugaite et al ("In search of robust measures of generalization") and proposed evaluation frameworks. How do the bounds vary as the depth changes? Width? As the learning rates vary?
- What insight would we gain from the bounds if they are tight in the cases tested? What insight into generalization in deep learning in general do evaluated CMI bounds provide?
- I wanted to confirm that the bounds for SGD (Adam) are for unperturbed-in-any-way classifiers, given that evaluated CMI is used. How was the averaging over draws of the data been done?

Other comments:

 - line 27 missing Neu et al COLT ’21, and Haghifam et al ’20 reference;

- Line 39: sounds like the measures are overfitting… but in any case, overfitting is a loaded word and here the authors should choose a different word precisely describing what they mean, in my opinion
- Eq. 25 bottom line, in P_{lambda | \tilde{Z}S} is it supposed to be  \tilde{Z}_S?

**Limitations:**

There is no explicit discussion of limitations. The assumptions are stated, though, but it might be good to add a short paragraph on the main assumptions that most theorems use, and that could be relaxed in future work.

**Strengths And Weaknesses:**

The connection between PAC-Bayesian bounds and information theoretic ones is one that is, by now, well understood. Indeed, Catoni's 2004 manuscript already spells out the relationship, but it's one that occasionally needs to be rediscovered or reiterated in the literature when we have our heads down and we forget to exploit this connection.

Is this work sufficiently novel/technical? The basic techniques for simultaneously deriving high probability and expected generalization bounds in the PAC-bayesian/info-theory setting have been provided, for example, in Hellstrom and Durisi (who the authors cite) and expanded recently by Grunwald, Steinke, and Zakyinthinou. And the MGF bounds for different g functions follow fairly trivially from Lemma 1, which itself appears in various forms in a number of papers (including Lever et al 2012 and some other works by Shawe-Taylor) used to derive basically any PAC-Bayes bound (just an expectation version of it).

Still, somehow the work on mutual information bounds has never gone beyond linear comparisons of the risk and empirical risk (leading to bounds on expected generalization error). The bounds displayed here for the case where empirical risk is zero are particularly interesting in my opinion, as similar bounds have been derived recently, but not from this perspective. And so I think in fact we had lost sight a bit about the role of a jointly convex comparator and it will be a useful observation for this subcommunity to reacquaint themselves with.

While the idea of combining these old insights from PAC-Bayes to mutual information bounds seems now obvious, the form of the resulting bounds (e.g., Theorem 4) is not what I expected, and so there seem to be some surprising consequences.

---

> ### Author Response · Authors · 2022-08-02
> **Response to Reviewer CMvk, part 1**
>
> We thank the reviewer for their constructive comments.
>
> ___________________________________________________________
> > _Is this work sufficiently novel/technical? The basic techniques for simultaneously deriving high probability and expected generalization bounds in the PAC-bayesian/info-theory setting have been provided, for example, in Hellstrom and Durisi (who the authors cite) and expanded recently by Grunwald, Steinke, and Zakyinthinou. And the MGF bounds for different g functions follow fairly trivially from Lemma 1, which itself appears in various forms in a number of papers (including Lever et al. 2012 and some other works by Shawe-Taylor) used to derive basically any PAC-Bayes bound (just an expectation version of it).
> Still, somehow the work on mutual information bounds has never gone beyond linear comparisons of the risk and empirical risk (leading to bounds on expected generalization error). The bounds displayed here for the case where empirical risk is zero are particularly interesting in my opinion, as similar bounds have been derived recently, but not from this perspective. And so I think in fact we had lost sight a bit about the role of a jointly convex comparator and it will be a useful observation for this subcommunity to reacquaint themselves with.
> While the idea of combining these old insights from PAC-Bayes to mutual information bounds seems now obvious, the form of the resulting bounds (e.g., Theorem 4) is not what I expected, and so there seem to be some surprising consequences._
>
>
> As the reviewer points out, the results in this paper largely build on powerful techniques available in the literature.
> However, from a technical perspective, the bounds that are given in terms of the binary KL divergence rely on the concentration inequality in Lemma 2, which pertains to non-identically distributed random variables and is, to the best of our knowledge, novel.
> Using this new concentration result, we avoid a logarithmic dependence on $n$ that would appear if we used standard techniques.
> Furthermore, previously obtained PAC-Bayesian results in terms of the binary KL divergence have not been derived for the e-CMI setting.
> Deriving such bounds for the e-CMI setting, which we do in this paper, requires careful handling of the random variables involved, leading to the unexpected form of Theorem 4 that you mention.
> This is in contrast to bounds based on the ordinary MI, which we discuss and extend in Appendix B.1.
> Finally, regarding the expressiveness of the e-CMI framework, previous work considers average bounds, while we take extra steps to bound the pointwise information measures that appear in our high-probability bounds.
> Thus, while Lemma 1 and its proof are straightforward, applying Lemma 1 for the specific convex functions and settings that we consider is non-trivial.
> Finally, the resulting bounds demonstrably improve upon previous work for some cases.
>
> ___________________________________________________________
> > _Questions:
> The comment under Theorem 8 says that the “resulting bounds essentially recover previously established generalization guarantees”. Are these previously established guarantees optimal? Are there lower bounds? What does “recover” here mean? Is the rate wrt the number of samples optimal in the minimax sense?_
>
> The generalization bounds in App. B.4, Eq. (117-118), and thus the bounds on the information terms in Theorem 8, are minimax optimal up to the logarithmic dependence on $n$, which can be removed for some cases (see, e.g., *Understanding Machine Learning: From Theory to Algorithms"*, Shai Shalev-Shwartz and Shai Ben-David, Theorem 29.3).
> By "recover", we mean that our information-theoretic bounds can be weakened to obtain known minimax generalization guarantees (up to a logarithmic term).
> We will clarify this in the revised version of the paper.

---

> > ### Author Response · Authors · 2022-08-02
> > **Response to Reviewer CMvk, part 2**
> >
> > ___________________________________________________________
> > > _The authors do not really explain what they hope to gain by performing the empirical evaluations, and what questions they are answering. It seems that the motivation is evaluating the numerical tightness of the bounds. Is that right? [...]
> > What insight would we gain from the bounds if they are tight in the cases tested? What insight into generalization in deep learning in general do evaluated CMI bounds provide?_
> >
> > The purpose of our numerical experiments is to probe the explanatory power of our bounds.
> > In this paper, we present upper bounds on the generalization gap in terms of the information that the losses of the learning algorithm on both training and test samples reveal about the selected data points in the CMI setting, given the entire supersample.
> > Our numerical results indicate that this quantity does capture the bulk of generalization performance for the settings that we consider.
> > Establishing this is the main goal of our empirical evaluations: if a bound closely follows the true test error---as we find that our bounds do for the settings that we study, which include experiments with partly randomized labels---it must capture the mechanism for generalization that is actually at play in the given setting.
> >
> > As we show in Section 4, the e-CMI framework is expressive enough to recover generalization guarantees in terms of classical complexity measures like the Natarajan dimension, and as shown by (Harutyunyan et al., 2021), it also captures some notions of algorithmic stability.
> > Further pursuing this line of inquiry to determine the concrete mechanism by which SGD-trained neural networks achieve small e-CMI is an interesting area for further study, but beyond the scope of this paper.
> >
> > ___________________________________________________________
> > > _It would have been nice to extend the evaluation in a few dimensions, beyond varying the number of training epochs and the number of training points. I encourage the authors to check out a paper by Dziugaite et al. ("In search of robust measures of generalization") and proposed evaluation frameworks. How do the bounds vary as the depth changes? Width? As the learning rates vary?_
> >
> > As you point out, and as is discussed in (Dziugaite et al., 2020), varying a wide variety of dimensions is necessary to determine if the bound has a robustly correct correlation with the true test error.
> > While this goes beyond our original purpose of probing the capacity of our bounds to capture the bulk of generalization in neural networks, it is of course an interesting experiment to run.
> > We have repeated the numerical experiments for SGD on binary MNIST with $n=75$ samples with varying depth, width, and learning rate, and the results can be found in Figure 5 of Appendix C in the updated supplementary material.
> > The reason for choosing $n=75$ is not only the lower computational cost, but also that the test loss is relatively high, so that variations in it are more noticeable.
> > The results show that our bounds display good correlation with the true test loss when varying depth, width, and learning rate.
> >
> > ___________________________________________________________
> > > _I wanted to confirm that the bounds for SGD (Adam) are for unperturbed-in-any-way classifiers, given that evaluated CMI is used. How was the averaging over draws of the data been done?_
> >
> > The SGD- and Adam-trained networks that we evaluate are indeed the unperturbed outputs of the training algorithms.
> > The resulting bounds would be vacuous if the ordinary CMI bounds were used, since the weights are continuous random variables.
> > On the contrary, the bounds are nonvacuous in the e-CMI setting, since the losses are discrete random variables.
> > The averaging over data is done by repeatedly sampling a subset of the full data set without replacement, running the training algorithm on this subset, and evaluating the test loss and bounds.
> >
> >
> > ___________________________________________________________
> > > _Limitations:
> > There is no explicit discussion of limitations. The assumptions are stated, though, but it might be good to add a short paragraph on the main assumptions that most theorems use, and that could be relaxed in future work._
> >
> > We will include such a paragraph in the revised version of the paper, where we will highlight the main theoretical assumptions, as well as clarify the purpose of the reported empirical evaluation and its limitations. We have also corrected the issues that are pointed out in "Other comments".

---

> > > ### Comment · Reviewer_CMvk · 2022-08-08
> > > **Thank you for clarifications**
> > >
> > > Thank you for further extending the experiments to different settings. I think it would be interesting to go beyond binary MNIST, and n=75, but the paper is worth publishing without these extensions.
> > >
> > > Regarding the evaluation of the bounds, I think it should be made clearer in the final version that the reported bounds and risk are for unperturbed predictors, and are coming from eCMI.

---

> > > > ### Author Response · Authors · 2022-08-08
> > > > **Thank you**
> > > >
> > > > Thank you for your detailed review, your helpful comments and suggestions, and for raising the score. In the revised version, we will extend these additional experiments to bigger data sets. We will also clarify that the reported bounds are based on the e-CMI of unperturbed predictors.

---

### Meta-Review · Area_Chair_zyym · 2022-08-23

**Recommendation:** Accept
**Confidence:** Certain

**Metareview:**

This paper presents a novel sequence of results providing generalization bounds for multi-class classification within PAC-Bayesian framework. These results extend and improve some of the existing bounds. Under some assumptions, these results give a tighter bound for multi-class classification with deep neural networks than previously existing results.

These results are of interest to theoretical ML community and a valuable contribution to the conference.

**Award:**

No

---

### Decision · Program_Chairs · 2022-09-14

Accept